# Socioeconomic deprivation and suicide in Appalachia: The use of three socioeconomic deprivation indices to explain county-level suicide rates

Erin D. Caswell[1,2]*, Summer D. Hartley[3], Caroline P. Groth[1], Mary Christensen[4], Ruchi Bhandari[1]

1 Department of Epidemiology and Biostatistics, School of Public Health, West Virginia University, Morgantown, WV, United States of America, 2 Health Affairs Institute, West Virginia University, Morgantown, WV, United States of America, 3 Hartley Health Solutions, Morgantown, WV, United States of America, 4 School of Social Work, West Virginia University, Morgantown, WV, United States of America

* erin.caswell@hsc.wvu.edu

**Data Availability Statement:** The data underlying the results presented in the study are publicly available from CDC Wonder (https://wonder.cdc.

## Abstract

### Objective

West Virginia's (WV) suicide rate is 50% higher than the national average and is the highest in the Appalachian Region. Appalachia has several social factors that have contributed to greater socioeconomic deprivation, a known contributor of suicide. Given WV's high prevalence of suicide and poverty, the current study aims to examine the relationship between socioeconomic deprivation and suicide rates in WV.

### Methods

The Townsend Deprivation Index (TDI), Social Deprivation Index (SDI), and Social Vulnerability Index (SVI) measured socioeconomic deprivation. Negative binomial regression models assessed the relationship between socioeconomic deprivation scores, individual index items, and suicide rates. Model comparisons evaluated the indices' ability to assess suicide rates. A backward selection strategy identified additional key items for examining suicide rates.

### Results

There was a significant increase in suicide rates for every 10% increase in TDI ($\beta = 0.04$; $p < 0.01$), SDI ($\beta = 0.03$; $p = 0.04$), and SVI scores ($\beta = 0.05$; $p < 0.01$). Household overcrowding and unemployment had a positive linear relationship with suicide in TDI ($\beta = 0.04$, $p = 0.02$; $\beta = 0.07$, $p = 0.01$), SDI ($\beta = 0.10$, $p = 0.02$; $\beta = 0.01$, $p<0.01$), and the SVI ($\beta = 0.10$, $p = 0.02$; $\beta = 0.03$, $p<0.01$). The backwards selection strategy identified additional key items included by the SVI when assessing suicide.

### Conclusion

Greater socioeconomic deprivation, measured by the TDI, SDI, and SVI, was significantly associated with higher suicide rates. Expanding unemployment benefits and increasing the

gov/mcd-icd10.html), the US Census Bureau ACS estimates (https://data.census.gov/), CDC/ATSDR (https://www.atsdr.cdc.gov/placeandhealth/svi/data_documentation_download.html), and Robert Graham Center (https://www.graham-center.org/maps-data-tools/social-deprivation-index.html).

**Funding:** The author(s) received no specific funding for this work.

**Competing interests:** The authors have declared that no competing interests exist.

availability of affordable housing, especially in rural areas, may be useful in reducing suicide rates. Our results suggest racial and ethnic minorities and adults living with a disability may benefit from targeted suicide prevention strategies.

## Introduction

Suicide is an urgent public health crisis in West Virginia, as the state's suicide rate is almost 50% higher than the national average and is the highest in the Appalachian Region [1]. The Appalachian Region is comprised of thirteen states across the eastern United States and is known to have several social factors, such as lack of social capital, increased substance use, lower educational attainment levels, barriers to transportation, and substandard housing characteristics, such as overcrowding [2, 3], that can be referred to in aggregate as socioeconomic deprivation. As the only state entirely within Appalachia, West Virginia reflects the profound socioeconomic deprivation that defines much of the region [4]. These factors collectively contribute to heightened vulnerability to suicide and other adverse outcomes—such as mental health disorders and substance abuse—by compounding stress stemmed from negative social and economic conditions. This vulnerability is particularly pronounced in rural areas, where individuals face greater isolation and have fewer resources to manage mental health challenges [5]. Socioeconomic deprivation, exacerbated by West Virginia's rural geography, has potentially contributed to a troubling rise in suicide rates.

Focusing research to West Virginia is critical not only because of its disproportionate impact on the state's residents but also because it underscores the broader challenges faced by the entire Appalachian Region [2–4, 6]. Consequently, Appalachian residents are 21% more likely to die by suicide than those in other parts of the United States [6], illustrating the unique vulnerabilities linked to the region's widespread socioeconomic deprivation. Despite these alarming statistics, there remains a significant gap in research on the relationship between suicide and socioeconomic deprivation in West Virginia and the broader Appalachian region. West Virginia's unique geographical and socioeconomic context makes it essential to understand local conditions, as studies from other United States regions show that suicide risk factors can vary considerably based on location. For example, factors like urban density and economic disparities in metropolitan areas may be more influential in broader United States contexts [7, 8]. Whereas, West Virginia's high rates of rural isolation, limited access to healthcare, and persistent economic hardship create distinct challenges. [9] Although, socioeconomic deprivation is often measured through various indices that assess the association between deprivation and suicide, has yet to be evaluated specifically within the Appalachian West Virginia population [10–13].

Given the region's unique social and economic landscape, there is a critical need to identify the specific factors that constitute socioeconomic deprivation in this area and determine their significant contributions to West Virginia's elevated suicide rates. Recent data underscores this urgency, as the state's age-adjusted suicide rates have risen in recent years, with 19.4 deaths per 100,000 people in 2020 to 20.6 in 2021 [1]. This estimate is markedly higher than the 2021 national age-adjusted suicide rate of 14.0 [14]. This rise in suicide rates has been attributed to the state's increased vulnerability to adverse socioeconomic factors, such as high poverty levels, which stood at 16.8% in 2021 compared to 12.8% nationally [15, 16]. Although often interconnected, it is crucial to differentiate between poverty, socioeconomic deprivation, and vulnerability for identifying associations with suicide. Poverty has been generally used to refer to a

lack of financial resources, typically measured by income [17]. Socioeconomic deprivation, however, is a broader concept that includes poverty but also captures additional material and social disadvantages such as limited access to education, substandard housing, inadequate transportation, and/or diminished social capital [18]. Vulnerability, on the other hand, reflects an increased susceptibility to adverse outcomes, such as suicide, due to the cumulative effects of socioeconomic deprivation [17, 19].

Different measures have been used to assess socioeconomic deprivation while examining its association with suicide. For example, socioeconomic deprivation has been assessed using measures such as unemployment, education level, and income [13] Studies in the United States have predominantly relied on income to measure poverty [20–23]. However, income as a standalone measure may not capture socioeconomic deprivation comprehensively and accurately [24]. By focusing solely on income, important contributors to social and material disadvantage are often overlooked, leading to an incomplete assessment of overall socioeconomic deprivation. Consequently, studies examining the association between poverty and suicide using a single poverty measure have shown varied results depending on the metric used to capture poverty. For example, a few studies found significant inverse relationships between poverty and suicide [25–28], while others found no association [28–30]. These discrepancies could stem not only from different poverty metrics used but also from other factors, including geographic variability, the methods employed to measure associations, and differing population characteristics. Hence the variability in findings reflects the multifaceted nature of socioeconomic deprivation and the context in which it is studied [31].

The complex nature of socioeconomic deprivation and related poverty suggests the need for a multidimensional measure. Moreover, such measure may accurately capture the true relationship between socioeconomic deprivation and suicide. Socioeconomic deprivation indices control for various poverty measures such as educational attainment, housing characteristics, availability of transportation, and income [32–34]. Using these indices to measure deprivation instead of a single poverty measure could provide additional insight into the contributors to suicide and socioeconomic deprivation in West Virginia.

Socioeconomic deprivation is experienced disproportionately among different West Virginian populations, distinctly among those with lower educational attainment levels, racial and ethnic minorities, and those residing in the southern, more rural areas of the state [4, 16, 35]. This disproportionate representation of socioeconomic deprivation across the state suggests the need to correctly identify areas of greatest concern for the implementation of targeted programs to promote a reduction of statewide suicide rates [36]. Measuring socioeconomic deprivation with a multidimensional deprivation index may more accurately and consistently identify high-need areas by accounting for the complexity of constructs when assessing its relationship with suicide. Given the high prevalence of suicide and socioeconomic deprivation in West Virginia, the current study aims to examine the relationship between socioeconomic deprivation and suicide rates in West Virginia.

## Methods

### Suicide estimates

Raw county-level suicide rates were calculated using data from the Centers for Disease Control and Prevention's (CDC) Wide-Ranging Online Data for Epidemiologic Research (WONDER) database for the years 2000–2010 [37]. Multiple Cause of Death ICD-10 codes, U03 (terrorism intentional suicide), X60-X84 (intentional self-harm), and Y87.0 (sequelae of intentional self-harm) identified suicide cases. Additional cases were those identified using the classification codes that represent accidental poisonings (X40-X43) and poisoning of undetermined intent

(Y10-Y15). There is a high prevalence of overdose mortality in West Virginia and Appalachia, and it is estimated that 80% of accidental poisonings and 90% of poisonings of undetermined intent are self-inflicted overdoses [38]. Therefore, these additional classification codes accounted for the intentional overdoses related to the ongoing drug crisis in West Virginia. Population estimates were also obtained from the WONDER database, and the analysis was restricted to adults aged 18 and older.

## Three socioeconomic deprivation indices

The relationship between socioeconomic deprivation and suicide rates in this study was assessed by three previously developed and validated indices: Townsend Deprivation Index, Social Deprivation Index, and Social Vulnerability Index [32, 34, 39]. The three indices were created using the most recent 2020 census 5-year American Community Survey (ACS) estimates. The indices cover various domains including socioeconomic status, housing type and transportation, household characteristics, and racial and ethnic minority status (S1 Table).

County-level Townsend Deprivation Index scores were manually created following the methodology outlined in the original publication [39]. Both unemployment and household overcrowding are non-normally distributed in the general population [40]. Therefore, following Townsend's methodology, these index items were first log-transformed. The log-transformed unemployment and household overcrowding items were standardized and then summed along with the county percentages of the population living in renter-occupied housing units and households without a car to produce the final Townsend Deprivation Index scores. For the Social Deprivation Index, a composite measure was created using demographic variables from the ACS. These variables were standardized and weighted based on factor analysis to include only those strongly associated with deprivation [32, 33]. To construct the Social Vulnerability Index, ACS variables were ranked for all United States census tracts, with each variable assigned a percentile rank. For most variables, higher ranks indicated greater vulnerability, except for per capita income, where higher values indicated less vulnerability. Percentile ranks were calculated for each variable and domain. Finally, an overall percentile rank was granted for each census tract [41]. County-level scores for the Social Deprivation Index and the Social Vulnerability Index were readily available from their host organization [32, 34].

## Statistical analysis

This study utilized an ecological study approach, and all analyses were performed using R Studio v.4.3.1 [42]. Maps were created from data obtained from the usmap R package and the ggthemes, ggplot, and tidyverse packages were used for data management and aesthetics [43–45]. Socioeconomic deprivation index scores were individually merged with county-level suicide rates by county location. In compliance with data suppression rules, counties with fewer than ten observations were excluded from our analyses [45]. Descriptive statistics for mortality characteristics, individual index items, and index scores were reported. Higher index scores indicated higher socioeconomic deprivation across all three indices. Negative binomial regression models were used to assess the relationship between county-level socioeconomic deprivation scores and suicide rates. Negative binomial regressions were chosen to account for the expected non-normal distribution of suicide rates and for over/under dispersion [46]. Consistent with suicidology research, models were adjusted for rural classification based on the 2013 Rural-Urban Continuum Codes (RUCC) [47, 48]. Analyses were performed on both unadjusted and adjusted models. Significance was set to .05. To standardize results across indices, regression results were reported to represent a 10% change in score. Standardization involved multiplying the respective index's score range by .1 and then multiplying the total by the

regression coefficient to demonstrate a 10% change. To evaluate each index's ability to assess county suicide rates in West Virginia, model comparisons were conducted using the model Akaike Information Criterion (AIC). To visually evaluate this performance, we created three maps using the fitted values generated from the adjusted models. A map depicting the raw county suicide rates was used for reference. Additional negative binomial regressions adjusted for rurality identified the index items that had a significant relationship with suicide. Finally, a backward selection modeling strategy identified key items for examining suicide rates in each index. The significance level was set to 0.10 to account for small sample sizes and identify trends among key items. The final data used for analysis has been made publicly available and can be found in the supplemental files (S1–S4 Files).

## Results

### Descriptive statistics

Of the 55 West Virginia counties, 34 (61.82%) fell under rural classification. Three counties (Calhoun, Gilmer, and Wirt) had less than ten suicide cases during the study period and were excluded from the analysis. An average of 127 (SD = 154.71) suicides per 100,000 people in West Virginia occurred from 2010 to 2020. While socioeconomic deprivation scores are not comparable between indices due to their distinct constructs and scales, which could distort the interpretation if standardized, the level of deprivation between counties can be interpreted from their respective county index scores [49]. The highest socioeconomic deprivation scores identified the counties with the greatest level of socioeconomic deprivation. Counties with the greatest deprivation had a Townsend Deprivation Index score of 3.30, a Social Deprivation Index score of 1.02, or a Social Vulnerability Index score of 11.44. Descriptive statistics of county suicide estimates, index scores, and index items are located in Table 1.

### Model results and comparisons

Models were adjusted for rural classification based on the 2013 RUCC [47]. A sensitivity analysis comparing adjusted and unadjusted model AICs revealed a substantial difference between models. Results from the adjusted negative binomial regressions (Table 2) demonstrate a significant increase in suicide rates for every 10% increase in Townsend Deprivation Index scores ($\beta = 0.04$; $p < 0.01$), Social Deprivation Index scores ($\beta = 0.03$; $p = 0.04$), and Social Vulnerability Index scores ($\beta = 0.05$; $p < 0.01$). The lowest model AIC was found in the Social Vulnerability Index (370.20), followed by the Townsend Deprivation Index (374.52) and Social Deprivation Index (381.29). Visual comparisons of maps created from adjusted model-fitted suicide rates further evaluated the indices' ability to assess county-level suicide rates (Fig 1). The reference map depicts the raw suicide rates obtained from CDC WONDER [37]. The map of suicide rates predicted by the Social Deprivation Index demonstrated the least variability in suicide rates among counties (30–43 deaths per 100,000 people).

### Examining individual index items

Results from the negative binomial regression analyses (Table 3) found a significant positive linear trend between the Social Vulnerability Index's poverty level item and suicide rates when adjusted for rurality. For every increase in the percentage of adults living below 150% of the federal poverty level, suicide rates increased ($\beta = 0.01$, $p < 0.01$). The poverty item in the Social Deprivation Index also had a significant relationship with suicide rates when adjusted for rurality ($\beta = 0.01$, $p = 0.02$). Household overcrowding and unemployment had a positive linear relationship with suicide in the Townsend Deprivation Index (overcrowding: $\beta = 0.04$

**Table 1. Descriptive statistics of county suicide, index scores, and index items.**

|  | Minimum | Maximum | Mean | SD |
|---|---|---|---|---|
| **Mortality Characteristics** |  |  |  |  |
| Suicide Deaths (n) | 12.00 | 857.00 | 127.24 | 154.71 |
| Total Population* (n) | 50150.00 | 291269.00 | 1635698.10 | 289816.50 |
| **Townsend Deprivation Index Score & Items** |  |  |  |  |
| Townsend Deprivation Index Score | -9.70 | 3.30 | -2.85 | 3.03 |
| Households Without Vehicle (%) | 2.11 | 16.28 | 8.06 | 2.57 |
| Living in Renter Occupied Housing (%) | 9.00 | 41.79 | 23.20 | 5.74 |
| Unemployed (%) | 2.89 | 15.01 | 7.43 | 3.04 |
| Overcrowded Households (%) | 0.15 | 4.15 | 1.27 | 0.72 |
| **Social Deprivation Index Score & Items** |  |  |  |  |
| Social Deprivation Index Score | -1.29 | 1.02 | 0.19 | 0.57 |
| Below 100% Poverty (%) | 5.84 | 32.81 | 17.51 | 5.27 |
| Aged 25 or Older w/Less Than 12 Years of Education (%) | 7.19 | 27.77 | 14.22 | 4.70 |
| Living in Renter Occupied Housing (%) | 9.00 | 41.79 | 23.20 | 5.74 |
| Overcrowded Households (%) | 0.15 | 4.15 | 1.27 | 0.72 |
| Single Parent Households (%) | 1.81 | 16.30 | 10.45 | 3.08 |
| Household Without Vehicle (%) | 2.11 | 16.28 | 8.06 | 2.57 |
| Unemployed (%) | 26.51 | 69.74 | 41.14 | 8.58 |
| **Social Vulnerability Index Score & Items** |  |  |  |  |
| Social Vulnerability Index Score | 5.11 | 11.44 | 7.82 | 1.45 |
| Below 150% Poverty (%) | 14.80 | 48.10 | 28.57 | 6.42 |
| Unemployment Rate | 2.90 | 15.00 | 7.44 | 3.03 |
| Housing Cost Burdened Households (%) | 11.50 | 29.40 | 18.33 | 4.01 |
| No High School Diploma (%) | 7.20 | 27.80 | 14.23 | 4.69 |
| Percent Uninsured (%) | 3.60 | 11.40 | 6.19 | 1.51 |
| Aged 65 and Older (%) | 12.70 | 28.10 | 21.29 | 2.69 |
| Aged 17 and Younger (%) | 15.00 | 23.30 | 19.80 | 1.82 |
| With Disability (%) | 11.90 | 33.10 | 20.59 | 5.05 |
| Single Parent Households (%) | 0.80 | 6.90 | 4.33 | 1.62 |
| Who Speak English "Less Than Well" (%) | 0.00 | 4.40 | 0.33 | 0.63 |
| Racial and Ethnic Minority (%) | 0.70 | 18.70 | 6.21 | 4.07 |
| Housing in Structures w/10 or More Units (%) | 0.10 | 15.50 | 2.86 | 3.21 |
| Mobile Homes (%) | 3.60 | 32.90 | 17.65 | 7.14 |
| Overcrowded Households (%) | 0.20 | 4.10 | 1.27 | 0.73 |
| Households Without Vehicle (%) | 2.10 | 16.30 | 8.06 | 4.10 |
| In Group Quarters (%) | 0.00 | 18.60 | 2.81 | 3.21 |

* Population estimates from combined 2010–2020 years retrieved from CDC's Multiple Cause of Death database

p = 0.02; unemployment: $\beta = 0.07$, $p = 0.01$), Social Deprivation Index (overcrowding: $\beta = 0.10$, $p = 0.02$; unemployment: $\beta = 0.01$, $p<0.01$), and the Social Vulnerability Index (overcrowding: $\beta = 0.10$, $p = 0.20$; unemployment: $\beta = 0.03$, $p<0.01$). Education was a significant item shared between the Social Deprivation Index and the Social Vulnerability Index. In the Social Deprivation Index, the percentage of those aged 25 years or older with less than 12 years of education exhibited a positive linear relationship with suicide rates ($\beta = 0.02$, $p<0.01$). Similarly, the Social Vulnerability Index showed a significant trend for the percentage of those without a high school diploma ($\beta = 0.02$, $p<0.01$). There were additional positive linear trends

**Table 2. Socioeconomic index model comparisons.**

|  | β | *p* | AIC |
|---|---|---|---|
| **TDI** |  |  |  |
| Unadjusted | 0.05 | *< 0.01** | 376.79 |
| Adjusted* | 0.04 | *<0.01** | 374.52 |
| **SDI** |  |  |  |
| Unadjusted | 0.02 | 0.10 | 385.37 |
| Adjusted* | 0.03 | 0.04* | 381.29 |
| **SVI** |  |  |  |
| Unadjusted | 0.05 | *< 0.01** | 374.99 |
| Adjusted* | 0.05 | *<0.01** | 370.20 |

* Models adjusted for rural classification based on 2013 Rural-Urban Continuum Codes (RUCC); Model coefficients represent a 10% change in score.

with suicide rates among the following Social Vulnerability Index items: percentage of housing cost-burdened households (β = 0.02, p = 0.01), percentage of adults with disability (β = 0.02, p<0.01), and percentage of ethnic and racial minority adults (β = 0.02, p = 0.01).

## Identifying key index items

A backward selection strategy conducted separately on the Townsend Deprivation Index, Social Deprivation Index, and Social Vulnerability Index identified additional key index items for assessing suicide rates. The results for the Townsend Deprivation Index and the Social Deprivation Index shown in S2 and S3 Tables. In the Social Vulnerability Index (Table 4), the first model included all significant items: unemployment rate, percent living below 150% of the federal poverty level, percent of overcrowded housing units, percent of housing cost-burdened households, percent of ethnic and racial minorities, and percent of those living with a disability. Following the backward selection strategy, items with the highest p-value were excluded from the models one at a time until all items remained significant at the 0.10 level. The final model, following this backward selection strategy, included the percentage of those without a high school diploma, racial and ethnic minorities, and those living with a disability. Although the percentage of individuals without a high school diploma had a p-value of 0.11, slightly above the threshold of .10, it was retained because its removal did not improve the Akaike Information Criterion (AIC) [48]. This inclusion ensured the best model fit [48]. When modeled together, these items showed a significant positive-linear (β = 3.04; p<0.01) relationship with county-level suicide rates. A similar strategy identified key Townsend Deprivation Index and Social Deprivation Index items. The Townsend Deprivation Index was comprised of only four items, two of which had a significant relationship with suicide. Results indicated that household overcrowding and unemployment had a significant positive-linear relationship with suicide (β = 3.90; p<0.01) when modeled together. The final model created for the Social Deprivation Index also had a significant positive linear relationship with suicide (β = 3.45; p<0.01) and included those aged 25 years or older with less than 12 years of education and overcrowded households.

## Discussion

Our findings, utilizing representative West Virginia suicide data, demonstrate that greater socioeconomic deprivation, as measured by the Townsend Deprivation Index, Social Deprivation Index, and Social Vulnerability Index, was significantly associated with higher county-

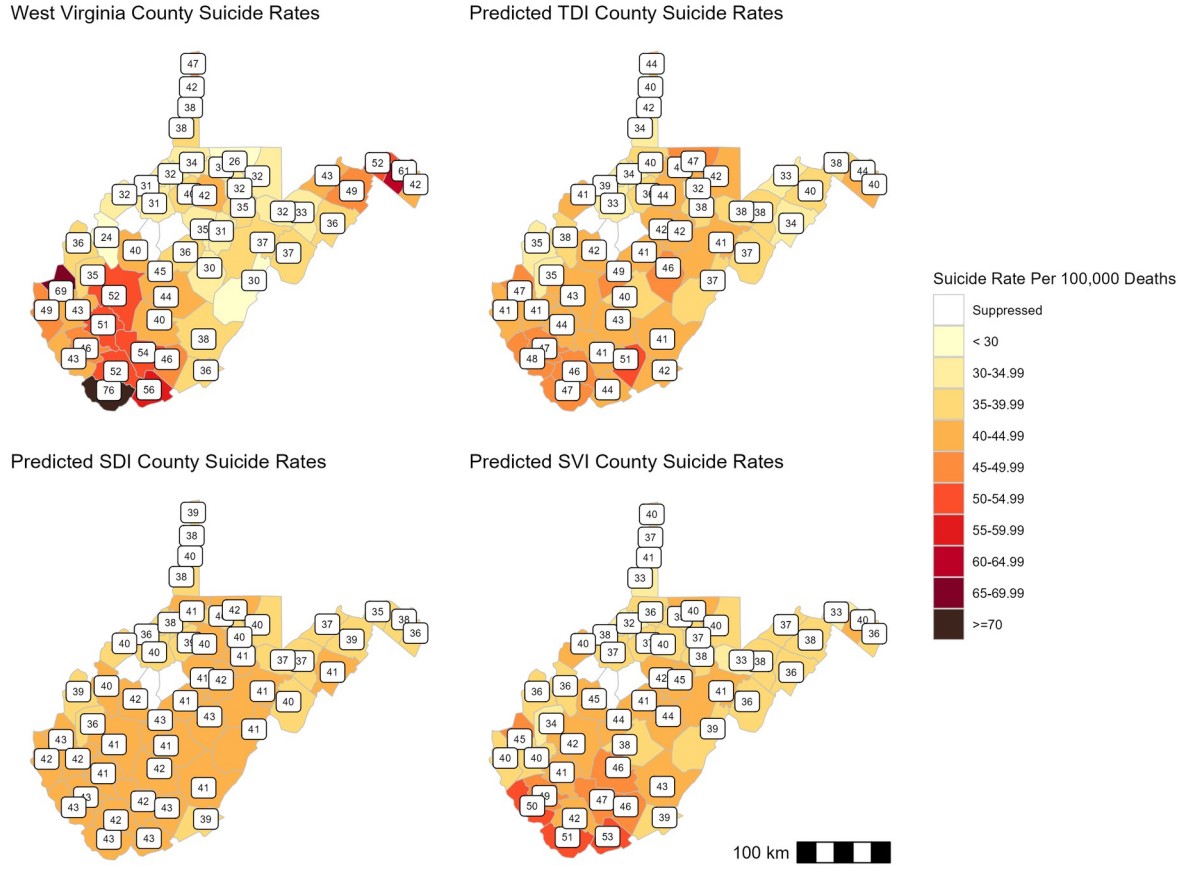

**Fig 1. Maps of index predicted suicide rates.**

level suicide rates when adjusting for rurality. These results are corroborated by previous studies that showed a significant statistical association between higher scores on these indices and higher suicide rates [10, 50].

We found the Townsend Deprivation Index and Social Vulnerability Index to have a similar statistical performance for assessing county-level suicide rates. However, there was a four-unit difference in model AIC, suggesting the Social Vulnerability Index may explain greater variability when examining suicide rates [51]. All three indices shared common items that had a significant association with suicide. Our results demonstrated that household overcrowding and unemployment were associated with increased suicide rates. Unemployment is a known risk factor for higher suicide rates [52–54], and its significance in Appalachia has been previously demonstrated [37]. However, little is known about the relationship between household overcrowding and suicide rates, especially in Appalachia, where adequate, affordable housing is in short supply and may contribute to substandard living conditions like overcrowding [55]. Notably, one study conducted in the region found higher rates of suicide among those with severe housing problems, which included overcrowded households [55], similar to our findings. Therefore, population-based suicide prevention interventions in Appalachia should focus on economic development that increases employment and improves accessibility to adequate and affordable housing, reducing the burden of socioeconomic deprivation in the population.

This study further identified significant items that were only present in the Social Vulnerability Index, in addition to unemployment and overcrowding. A backward selection strategy

**Table 3. Individual index items: Negative binomial regression results.**

| | Estimate | p-value |
|---|---|---|
| **Townsend Deprivation Index Items** | | |
| Households Without Vehicle (%) | 0.04 | 0.17 |
| Living in Renter Occupied Housing (%) | 0.03 | 0.37 |
| Unemployed (%) | 0.07 | 0.01* |
| Overcrowded Households (%) | 0.04 | 0.02* |
| **Social Deprivation Index Items** | | |
| Below 100% Poverty (%) | 0.01 | 0.02* |
| Aged 25 or older w/Less Than 12 Years of Education (%) | 0.02 | <0.01* |
| Living in Renter Occupied Housing (%) | 0.01 | 0.36 |
| Overcrowded Households (%) | 0.10 | 0.02* |
| Single Parent Households (%) | 0.02 | 0.13 |
| Household Without Vehicle (%) | 0.02 | 0.17 |
| Unemployed (%) | 0.01 | <0.01* |
| **Social Vulnerability Index Items** | | |
| Below 150% Poverty (%) | 0.01 | <0.01* |
| Unemployment Rate | 0.03 | <0.01* |
| Housing Cost Burdened Households (%) | 0.02 | 0.01* |
| No High School Diploma (%) | 0.02 | <0.01* |
| Percent Uninsured (%) | -0.02 | 0.40 |
| Aged 65 or Older (%) | 0.00 | 0.95 |
| Aged 17 or Younger (%) | 0.02 | 0.19 |
| Living with Disability (%) | 0.02 | <0.01* |
| Single Parent Households (%) | 0.02 | 0.27 |
| Who Speak English "Less Than Well" (%) | -0.01 | 0.89 |
| Racial and Ethnic Minority (%) | 0.02 | 0.01* |
| Housing in Structures w/10 or More Units (%) | -0.01 | 0.38 |
| Mobile Homes (%) | 0.01 | 0.16 |
| Overcrowded Households (%) | 0.10 | 0.02* |
| Households Without Vehicle (%) | 0.02 | 0.17 |
| In Group Quarters (%) | 0.00 | 0.71 |

Associations between index items and county suicide rates adjusted for rural classification based on the 2013 Rural-Urban Continuum Codes (RUCC); To account for small sample sizes, significance was set at alpha = 0.1

identified the percentage of those with less than a high school education, those living with a disability, and racial and ethnic minorities as significant items when assessing their relationship with suicide. Results showed that an increase in any one of these items was associated with higher rates of suicide. In rural Appalachia and WV, educational attainment levels are especially low and contribute to socioeconomic disparities in the region [56]. The greater rates of suicide among those with less education implies the need to strategically communicate public benefit resources and suicide prevention campaigns to enhance their effectiveness in this population.

Another key Social Vulnerability Index item significantly associated with high suicide rates was disability. Our data indicated a high prevalence of adults (almost one-fifth) living with a disability. One type of disability is chronic pain. Chronic pain is prominent in West Virginia and the entire Appalachian region [57] and has a strong association with suicide [58, 59]. In Appalachia, prescription opioids were commonly prescribed by providers for chronic pain

**Table 4. Results from backward selection strategy: Social Vulnerability Index items.**

| Variable | Model 1 AIC = 366.62 β | p | Model 2 AIC = 364.64 β | p | Model 3 AIC = 362.86 β | p | Model 4 AIC = 361.96 β | p | Model 5 AIC = 360.95 β | p |
|---|---|---|---|---|---|---|---|---|---|---|
| Intercept | 2.90 | <0.01* | 2.90 | <0.01* | 2.91 | <0.01* | 2.93 | <0.01* | 3.04 | <0.01* |
| Rurality | -0.12 | 0.04* | -0.12 | 0.04* | -0.13 | 0.03* | -0.14 | 0.02* | -0.15 | 0.01* |
| No Highschool Diploma | 0.01 | 0.14 | 0.01 | 0.14 | 0.01 | 0.14 | 0.01 | 0.15 | 0.01 | 0.11 |
| Living with Disability | 0.02 | 0.03* | 0.02 | 0.02* | 0.02 | 0.02* | 0.02 | <0.01* | 0.02 | <0.01* |
| Racial and Ethnic Minority | 0.02 | <0.01* | 0.02 | <0.01* | 0.02 | <0.01* | 0.03 | <0.01* | 0.03 | <0.01* |
| Housing Cost Burdened Households | 0.01 | 0.23 | 0.01 | 0.24 | 0.01 | 0.26 | 0.01 | 0.32 | — | — |
| Overcrowded Households | 0.03 | 0.42 | 0.03 | 0.42 | 0.04 | 0.29 | — | — | — | — |
| Below 150% Poverty | 0.00 | 0.62 | 0.00 | 0.63 | — | — | — | — | — | — |
| Unemployment Rate | 0.00 | 0.89 | — | — | — | — | — | — | — | — |

Models adjusted for rural classification based on the 2013 Rural-Urban Continuum Codes (RUCC); To account for small sample sizes, *significance was set at alpha = 0.10

management. The overprescription of opioids is a root cause of the opioid crisis [60]. The opioid crisis has had several negative socioeconomic effects that have led to a high prevalence of overdose mortality and hindered the economic development of the Appalachian region [57]. Thus, our results suggest that highly concentrated areas of people living with disabilities may negatively influence socioeconomic deprivation and suicide rates, given the known contributions of chronic pain and related opioid crisis. Therefore, individual-based suicide interventions may be more effective when used in combination with chronic pain management and overdose prevention strategies when deemed appropriate through clinical assessment.

Lastly, the percentage of racial and ethnic minorities was also a key Social Vulnerability Index item significantly associated with high suicide rates. Although the mean county prevalence of racial and ethnic minority adults was low (6%), the variability in this subgroup was high (SD = 4.07). This variability could explain how higher concentrations of racial/ethnic minority populations may influence county-level suicide rates. Suicide rates are already disproportionately higher among this population [61]. Additionally, Appalachian racial/ethnic minorities experience greater socioeconomic deprivation [61]. They are more likely to live in overcrowded households, have a lower annual income, and have less access to quality health care [62, 63]. These underlying factors may be contributing to the disproportionately higher rates of suicide among racial/ethnic minority populations. Hence, the results from our study suggest that targeted interventions aimed at reducing socioeconomic disparities experienced in this group may also help lower suicide rates.

In addition to identifying the several key socioeconomic deprivation index items associated with suicide, rurality was found to have a significant role in the relationship between socioeconomic deprivation and suicide. This finding is consistent with prior findings that demonstrated higher rates of suicide in rural areas [7, 36, 64, 65]. These results suggest the need to focus suicide prevention efforts in rural areas, specifically among adults who are racial and ethnic minorities, have less education, are unemployed, live in overcrowded households, live with a disability. In West Virginia and the Appalachian Region, there are unique geographic barriers exacerbated by rurality that limit the accessibility to affordable housing, health care, job opportunities, and public health resource programs [66] and may contribute to greater socioeconomic deprivation and suicide.

## Strengths and limitations

This is the first study conducted in Appalachia to explore the relationship between socioeconomic deprivation (measured by a socioeconomic deprivation index) and suicide rates. In addition, exploring the individual items comprising each index may provide valuable insight for future research when examining Appalachian suicide rates. Despite the value of our findings, there are several limitations to note. (1) We complied with data suppression rules and suppressed data for counties that had ten or fewer suicides. This resulted in a small number of observations in several counties. Since index calculations used data only from 2020, there may be changes in items across time that were unaccounted for by the indices. (2) The ICD-10 codes used to identify cases of suicide differ from those used by the West Virginia Department of Health (WV DOH). The WV DOH excludes ICD-10 codes that represent drug intoxication. However, the inclusion of the additional ICD-10 codes may more accurately represent suicide in the Appalachian context because of the known socioeconomic effects of the opioid crisis in the region. Thus, we accounted for intentional overdoses to produce results representative of Appalachia [38]. (3) Our results only suggest correlations between indices, their items, and suicide. A non-linear relationship was not explored in our analyses. (4) Multicollinearity could be present in our analyses but did not appear to be substantial based on the variance inflation factors (VIF), with the highest VIF observed being 1.13 in our analyses. (5) Although tests indicated that there was no significant overdispersion in our data, we selected the negative binomial regression model to provide a more conservative approach and ensure robustness in our analysis [46]. (6) Lastly, this ecological study used data representative of West Virginia, and results may not be generalizable to counties outside of West Virginia or Appalachia.

## Conclusion

The Townsend Deprivation Index, Social Deprivation Index, and Social Vulnerability Index are appropriate measures of socioeconomic deprivation when assessing county-level suicide rates. All three indices had a significant relationship with suicide and include key items for assessing suicide rates. Future directions should evaluate the psychometric properties of these indices to create a socioeconomic deprivation index most appropriate for assessing suicide rates. Our results suggest that expanding unemployment benefits and increasing the availability of affordable housing may be particularly useful in reducing suicide rates. In addition, prevention strategies should target racial and ethnic minorities, adults living with a disability, and those with less education. It may be particularly relevant to add concurrent treatment of chronic pain management and overdose prevention to suicide prevention strategies when clinically appropriate, given the known contribution of the opioid crisis in the Appalachian Region.

## Supporting information

**S1 Table. Socioeconomic Deprivation Index domains & items.**
(DOCX)

**S2 Table. Results from backward selection strategy: Townsend Deprivation Index Items.**
(DOCX)

**S3 Table. Results from backward selection strategy: Social Deprivation Index items.**
(DOCX)

**S1 File. SDI final data with codebook.**
(XLSX)

**S2 File. SVI final data with codebook.**
(XLSX)

**S3 File. TDI final data with codebook.**
(XLSX)

**S4 File. WV suicide rates.**
(XLSX)

## Author Contributions

**Conceptualization:** Erin D. Caswell.

**Formal analysis:** Erin D. Caswell.

**Methodology:** Erin D. Caswell.

**Supervision:** Ruchi Bhandari.

**Writing – original draft:** Erin D. Caswell.

**Writing – review & editing:** Erin D. Caswell, Summer D. Hartley, Caroline P. Groth, Mary Christensen, Ruchi Bhandari.

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
