## [Decision Letter · Decision Letter 0]

20 Aug 2024

PONE-D-24-10230Socioeconomic deprivation and suicide in Appalachia: The use of three socioeconomic deprivation indices to explain county-level suicide ratesPLOS ONE

Dear Dr. Caswell,

Thank you for submitting your manuscript to PLOS ONE. After careful consideration, we feel that it has merit but does not fully meet PLOS ONE’s publication criteria as it currently stands. Therefore, we invite you to submit a revised version of the manuscript that addresses the points raised during the review process.

**ACADEMIC EDITOR: **

Dear author,

I hope this message finds you well. I apologize for the delay in sending you the feedback on your manuscript. Unfortunately, we encountered challenges in securing additional reviewers for your article, which resulted in the delay.

We have received a detailed review from one of the reviewers, who have suggested a major revision to ensure the manuscript meets the standards of our journal. Please carefully address the concerns raised by the reviewer. If you have any questions or require clarification on any of the points, feel free to reach out, and I will be happy to assist you.

Thank you for your understanding and patience throughout this process. We appreciate your commitment to the quality and integrity of academic publishing.

We look forward to receiving your revised manuscript.

Kind regards,

Angela Mendes Freitas

Academic Editor

PLOS ONE

Journal Requirements:

Reviewers' comments:

Reviewer's Responses to Questions

**Comments to the Author**

1. Is the manuscript technically sound, and do the data support the conclusions?

Reviewer #1: Partly

2. Has the statistical analysis been performed appropriately and rigorously? 

Reviewer #1: I Don't Know

3. Have the authors made all data underlying the findings in their manuscript fully available?

Reviewer #1: No

4. Is the manuscript presented in an intelligible fashion and written in standard English?

Reviewer #1: Yes

5. Review Comments to the Author

Reviewer #1: General comments

1) The study aimed to examine the relationship between socioeconomic deprivation and suicide rates in West Virginia, given the local high prevalence of both suicide and socioeconomic deprivation.

2) The authors used three well-known deprivation/vulnerability indices to assess their relationship with suicide rates. The three selected indices were: i) the Townsend Deprivation Index (TDI), ii) Social Deprivation Index (SDI), and iii) Social Vulnerability Index (SVI). Negative binomial regression models assessed the relationship between socioeconomic deprivation scores, individual index variables, and suicide rates. Model comparisons were conducted using the Akaike Information Criterion (AIC). A backward selection identified key variables for examining suicide rates. The models were adjusted for rurality.

3) The study does not appear to have ethical problems. The minimum number of the population per county was 50150.00 and of the 55 West Virginia counties, 3 counties had less than 10 suicide cases during the study period and were excluded from the analysis. Although the authors mentioned that the study follows the rules for data suppression in areas with few cases, they did not present the reference used to support this choice.

4) The authors did not provide the dataset used for the study. They only reported that the data underlying the results presented in the study are publicly available through different websites. Ideally, the data should be provided as part of the manuscript or its supporting information, or deposited in a single public repository.

Major revision

1) In the introduction, some main ideas appear repeatedly, mainly regarding the local evidence, while the main findings and gaps of similar studies developed for other regions of the world were not explored in depth. The topic has already been extensively studied for other regions, and the introduction should be aligned with current specific literature. In this sense, the local evidence could be reorganized into one or two paragraphs and more robust evidence on the relationship between the suicide rates and socioeconomic deprivation from studies developed for other regions should be considered in the introduction.

2) In the following lines, the authors concluded that the differences found in the associations between poverty and suicide by previous studies were due to the poverty measure. However, many other factors could be influencing it (e.g., the geographic area, the method used to measure the association, etc.). Thus, it is necessary to explain how these studies are relatable and if there is evidence for the same area and using the same methods, for example.

Lines 74-77. “Therefore, studies examining the association between poverty and suicide using a single poverty measure have shown varied results depending on the metric used to capture poverty. For example, a few studies found significant inverse relationships between poverty and suicide [20-23], while others found no association [23-25]”.

3) In general, along the introduction, the authors often referred about the potential effects of “poverty” and “socioeconomic deprivation” on suicide. The attempt to distinguish these concepts was not clear (e.g., lines 70-74). It is necessary to define these two concepts, explain how they are related and rewrite the introduction accordingly, to give a better understanding of the problem. I think it is also important to define and explain the relationship between these two concepts and “vulnerability”, since one of the selected indices were the Social Vulnerability Index (SVI).

4) In the section “Identifying Key Index Items”, lines 213-214, the authors mentioned that the final model, following the backward selection strategy, included the percentage of those without a high school diploma and other variables. According to the methods, items with the highest p-value should be excluded from the models one at a time until all items remained significant at the 0.10 level (lines 211-212). In this sense, the inclusion of the percentage of those without a high school diploma seems a mistake. The p-value of this variable (0.11) is quite close to the significant level of reference, but its inclusion needs to be justified.

Minor revision

1) It is necessary to mention the area that is being compared to the Appalachian Region in the following lines:

Lines 48-49. “The Appalachian Region has an urgency to reduce suicide rates, as its residents are 21% more likely to die by suicide”.

2) There is duplicate information in the following lines:

Lines 101-103. “Additional cases were those identified using the classification codes that represent accidental poisonings 101 (X40-X43) and poisoning of undetermined intent (Y10-Y15). X40-X43 (accidental poisoning and Y10- 102 Y15 (poisoning of undetermined intent)”.

3) In the “Suicide Estimates”, it is important to mention the source of the population estimates used to calculate the suicide rates. Only in Table 1 it was possible to conclude that it was obtained from the same database as suicide cases.

4) In the “Three Socioeconomic Deprivation Indices”, it is important to cite the first publications of the methodologies used to calculate the selected indices. As the different deprivation/vulnerability indices are central elements of the study and different methods can produce different results of deprivation, a brief description of how the scores of these indices are obtained is also needed in this section.

5) In the “Statistical analysis”, lines 125-126, it is important to cite the reference used for the data suppression.

6) In the “Statistical analysis”, lines 130-131, it is important to cite the references that justified the choice for the model (negative binomial model). I am not sure if the negative binomial regression can handle underdispersion, specifically. Furthermore, did the data present overdispersion or underdispersion? That wasn't clear.

7) In the “Statistical analysis”, lines 131-133, it is important to cite the references of suicide research used to justify why the models were adjusted for rurality.

8) In the “Descriptive Statistics”, the authors mention that the socioeconomic deprivation scores are not comparable between indices (lines 149-150). The interpretation of the results could be facilitated by transposing the scores of the 3 indices into the same scale of values.

9) In the lines 160-162, the following information does not coincide with the data shown in Table 2: “The lowest model AIC was found in the Social Vulnerability Index (370.20), followed by the Townsend Deprivation Index (374.52) and Social Deprivation Index (374.52)”.

10) The scale of the map must be inserted in Figure 1.

11) In lines 165-166, it seems better to exclude the following information: “When compared to the reference map”. The conclusion does not seem to result from a comparison with the reference map.

“When compared to the reference map, the map of suicide rates predicted by the Social Deprivation Index demonstrated the least variability in suicide rates among counties (30-43 deaths per 100,000 people)”.

12) In the section “Examining Individual Index Items”, the authors forgot to mention the significant value observed for the “Aged 25 or older with Less Than 12 Years of Education (%)” variable, which composes the SDI. A variable related to education was presented in 2 out of the 3 indices (the SDI and SVI). In both cases, the variable related to education presented a significant value (Table 3), but it was ignored in the text, in which the authors only mentioned the other significant variables.

13) In the section “Identifying Key Index Items” (lines 208-211), the authors forgot to mention the percentage of those without a high school diploma (β=0.02, p<0.01).

14) In the discussion (lines 300-301), the authors mentioned that multicollinearity could be present in the analyses but did not appear to be substantial based on the variance inflation factors. It is recommended to show the maximum value observed for multicollinearity.

15) In Supporting Information (S1_Table), it is recommended to show which age group was considered to calculate the unemployment rate of the SVI.

6. PLOS authors have the option to publish the peer review history of their article (what does this mean?). If published, this will include your full peer review and any attached files.

Reviewer #1: No

---

## [Author Response · Author response to Decision Letter 0]

4 Sep 2024

Reviewer #1: General comments

1) The study aimed to examine the relationship between socioeconomic deprivation and suicide rates in West Virginia, given the local high prevalence of both suicide and socioeconomic deprivation.

Response: We have followed the advice of the reviewer and have restructured the introduction to align clearly with this aim. 

2) The authors used three well-known deprivation/vulnerability indices to assess their relationship with suicide rates. The three selected indices were: the Townsend Deprivation Index (TDI), ii) Social Deprivation Index (SDI), and iii) Social Vulnerability Index (SVI). Negative binomial regression models assessed the relationship between socioeconomic deprivation scores, individual index variables, and suicide rates. Model comparisons were conducted using the Akaike Information Criterion (AIC). A backward selection identified key variables for examining suicide rates. The models were adjusted for rurality.

Response: The methodology may have been unclear in the original manuscript. Based on the reviewer comments, we clarified our methodology and provided additional details with supporting references in the revised manuscript. 

3) The study does not appear to have ethical problems. The minimum number of the population per county was 50150.00 and of the 55 West Virginia counties, 3 counties had less than 10 suicide cases during the study period and were excluded from the analysis. Although the authors mentioned that the study follows the rules for data suppression in areas with few cases, they did not present the reference used to support this choice.

Response: We have now included this reference and have described our choices below in the detailed response letter. 

4) The authors did not provide the dataset used for the study. They only reported that the data underlying the results presented in the study are publicly available through different websites. Ideally, the data should be provided as part of the manuscript or its supporting information, or deposited in a single public repository.

Response: We have now included the clean data we used for the final analyses along with this manuscript. 

Comments from Reviewer #1:

Major revisions

1) In the introduction, some main ideas appear repeatedly, mainly regarding the local evidence, while the main findings and gaps of similar studies developed for other regions of the world were not explored in depth. The topic has already been extensively studied for other regions, and the introduction should be aligned with current specific literature. In this sense, the local evidence could be reorganized into one or two paragraphs and more robust evidence on the relationship between the suicide rates and socioeconomic deprivation from studies developed for other regions should be considered in the introduction.

Response: 

Thank you for your insightful feedback. We appreciate your suggestions on improving the depth and breadth of the introduction. In response, based on a robust review of literature, we have restructured the introduction to focus on the specific context of Appalachia, aligning with the aim of the study. This revision highlights the distinctive challenges faced by West Virginia in comparison to broader contexts, such as those observed in other areas of the United States. We, thus, aim to provide a clearer understanding of how socioeconomic deprivation impacts suicide rates specifically in West Virginia.

Pages 3-4 Lines 48-74. 

“……The Appalachian Region is comprised of thirteen states across the eastern United States and the region is known to have several social factors, such as lack of social capital, increased substance use, lower educational attainment levels, barriers to transportation, and substandard housing characteristics, such as overcrowding [2,3], that can be referred to in aggregate as socioeconomic deprivation. As the only state entirely within Appalachia, West Virginia reflects the profound socioeconomic deprivation that defines much of the region. [4] These factors collectively contribute to heightened vulnerability to suicide and other adverse outcomes—such as mental health disorders and substance abuse—by compounding stress stemmed from negative social and economic conditions. [5] This vulnerability is particularly pronounced in rural areas, where individuals face greater isolation and have fewer resources to manage mental health challenges. [6] Socioeconomic deprivation, exacerbated by West Virginia’s rural geography, has potentially contributed to a troubling rise in suicide rates.

Focusing research to West Virginia is critical not only because of its disproportionate impact on the state's residents but also because it underscores the broader challenges faced by the entire Appalachian Region. [2-4, 7] Consequently, Appalachian residents are 21% more likely to die by suicide than those in other parts of the United States [7], illustrating the vulnerabilities linked to the region’s widespread socioeconomic deprivation. West Virginia's unique geographical and socioeconomic context makes it essential to understand local conditions, as studies from other United States regions show that suicide risk factors can vary considerably based on location. For example, factors like urban density and economic disparities in metropolitan areas may be more influential in broader United States contexts. [8-9] Whereas, West Virginia’s high rates of rural isolation, limited access to healthcare, and persistent economic hardship create distinct challenges. [6] Although, socioeconomic deprivation is often measured through various indices that assess the association between deprivation and suicide, has yet to be evaluated specifically within the Appalachian West Virginia population. [10-13]

Given the region's unique social and economic landscape, there is a critical need to identify the specific factors that constitute socioeconomic deprivation in this area and determine their significant contributions to West Virginia’s elevated suicide rates…….”

References: 

Palomin A, Takishima-Lacasa J, Selby-Nelson E, Mercado A. Challenges and Ethical Implications in Rural Community Mental Health: The Role of Mental Health Providers. Community Ment Health J. 2023;59(8):1442-51.

Rogerson P, Yang J, Bagchi-Sen S. Recent geographic patterns in suicide in the United States. GeoJournal. 2024;89(1):19.

Mukherjee S, Wei Z. Suicide disparities across metropolitan areas in the US: A comparative assessment of socio-environmental factors using a data-driven predictive approach. PLoS One. 2021;16(11):e0258824.

Marshall J, Thomas, L., Lane, N.M., Holmes, G. M., Arcury, T. A., Randolph, R., Silberman, P., Holding, W., Villamil, l., Thomas, S., Lane, M., Latus, J., Rodgers, J., Ivey K. Health Disparities in Appalachia. 2017.

Mezzina R, Gopikumar V, Jenkins J, Saraceno B, Sashidharan SP. Social Vulnerability and Mental Health Inequalities in the "Syndemic": Call for Action. Front Psychiatry. 2022;13:894370.

2) In the following lines, the authors concluded that the differences found in the associations between poverty and suicide by previous studies were due to the poverty measure. However, many other factors could be influencing it (e.g., the geographic area, the method used to measure the association, etc.). Thus, it is necessary to explain how these studies are relatable and if there is evidence for the same area and using the same methods, for example.

Lines 74-77. “Therefore, studies examining the association between poverty and suicide using a single poverty measure have shown varied results depending on the metric used to capture poverty. For example, a few studies found significant inverse relationships between poverty and suicide [20-23], while others found no association [23-25]”.

Response: 

Thank you for pointing out the need for a more nuanced discussion regarding the variability in findings from previous studies. To address this concern, we have revised the relevant section to include:

Pages 4-5 Lines 94-101. 

“Therefore, studies examining the association between poverty and suicide using a single poverty measure have shown varied results depending on the metric used to capture poverty. For example, a few studies found significant inverse relationships between poverty and suicide [23-26], while others found no association. [26-28] These discrepancies could stem not only from different poverty metrics used but also from other factors, including geographic variability, the methods employed to measure associations, and differing population characteristics. Hence the variability in findings reflects the multifaceted nature of socioeconomic deprivation and the context in which it is studied. [29]”

References: 

Lemmi V, Bantjes J, Coast E, Channer K, Leone T, McDaid D, et al. Suicide and poverty in low-income and middle-income countries: a systematic review. The Lancet Psychiatry. 2016;3(8):774-83.

3) In general, along the introduction, the authors often referred about the potential effects of “poverty” and “socioeconomic deprivation” on suicide. The attempt to distinguish these concepts was not clear (e.g., lines 70-74). It is necessary to define these two concepts, explain how they are related and rewrite the introduction accordingly, to give a better understanding of the problem. I think it is also important to define and explain the relationship between these two concepts and “vulnerability”, since one of the selected indices were the Social Vulnerability Index (SVI).

Response: 

Thank you for highlighting the need for clarity in distinguishing between “poverty,” “socioeconomic deprivation,” and “vulnerability.” In response to your comment, we have revised the introduction to explicitly define and differentiate these concepts and explain their interrelationships. 

Page 4. Lines 79-86. 

“Although often interconnected, it is crucial to differentiate between “poverty”, "socioeconomic deprivation," and "vulnerability" for identifying associations with suicide. Poverty refers specifically to a lack of financial resources, while socioeconomic deprivation encompasses a wider range of hardships, such as limited access to education, substandard housing, inadequate transportation, and diminished social capital. Unlike poverty, which focuses on income, socioeconomic deprivation provides a more holistic view by capturing multiple dimensions of material and social disadvantage. Vulnerability, on the other hand, reflects an increased susceptibility to adverse outcomes, such as suicide, due to the cumulative effects of socioeconomic deprivation. [17]”

References:

Biswas S, Nautiyal S. A review of socio-economic vulnerability: The emergence of its theoretical concepts, models and methodologies. Natural Hazards Research. 2023;3(3):563-71.

4) In the section “Identifying Key Index Items”, lines 213-214, the authors mentioned that the final model, following the backward selection strategy, included the percentage of those without a high school diploma and other variables. According to the methods, items with the highest p-value should be excluded from the models one at a time until all items remained significant at the 0.10 level (lines 211-212). In this sense, the inclusion of the percentage of those without a high school diploma seems a mistake. The p-value of this variable (0.11) is quite close to the significant level of reference, but its inclusion needs to be justified.

Response: Thank you for your feedback. We acknowledge that this approach may require further clarification. When conducting the backward selection, we considered not only the significance of each item but also the AIC. For instance, after removing education from model 5, the AIC increased, indicating a worse model fit. Therefore, education was retained in the final model to ensure the best fit. We have now included the following rationale for including the percentage of individuals without a high school diploma in the model within the results and methods sections.

Page 8 Lines 174-177

“Finally, a backward selection modeling strategy accounting for overall model improvement in AIC identified key items for examining suicide rates in each index. The significance level was set to 0.10 to account for small sample sizes and identify trends among key items. If removing an item increased model AIC significantly, the prior model was selected as the preferred model.”

Page 13. Lines 249-252

“Although the percentage of individuals without a high school diploma had a p-value of 0.11, slightly above the threshold of .10, it was retained because its removal did not improve the Akaike Information Criterion (AIC). This inclusion ensured the best model fit. [50]”

Reference: 

Chowdhury MZI, Turin TC. Variable selection strategies and its importance in clinical prediction modelling. Fam Med Community Health. 2020;8(1):e000262.

Minor revision

1) It is necessary to mention the area that is being compared to the Appalachian Region in the following lines:

Lines 48-49. “The Appalachian Region has an urgency to reduce suicide rates, as its residents are 21% more likely to die by suicide”.

Response: Thank you for your comment. I agree the original wording was unclear. I have revised the sentence for clarity:

Page 3 Lines 62-63

“Appalachian residents are 21% more likely to die by suicide than those in other parts of the United States [7].”

2) There is duplicate information in the following lines:

Lines 101-103. “Additional cases were those identified using the classification codes that represent accidental poisonings 101 (X40-X43) and poisoning of undetermined intent (Y10-Y15). X40-X43 (accidental poisoning and Y10- 102 Y15 (poisoning of undetermined intent)”.

Response: Thank you for highlighting this duplication. The repeated line, “X40-X43 (accidental poisoning) and Y10-Y15 (poisoning of undetermined intent),” has been removed.

3) In the “Suicide Estimates”, it is important to mention the source of the population estimates used to calculate the suicide rates. Only in Table 1 it was possible to conclude that it was obtained from the same database as suicide cases.

Response: Thank you for your helpful suggestion. I have revised the final sentence in the “Suicide Estimates” section to clarify the source of the population estimates:

Page 5. Line 129-130.

“Population estimates were also obtained from the WONDER database, and the analysis was restricted to adults aged 18 and older.”

4) In the “Three Socioeconomic Deprivation Indices”, it is important to cite the first publications of the methodologies used to calculate the selected indices. As the different deprivation/vulnerability indices are central elements of the study and different methods can produce different results of deprivation, a brief description of how the scores of these indices are obtained is also needed in this section.

Thank you for your valuable feedback. In response to your suggestion, we have cited the original publications for the methodologies used to calculate each of the three socioeconomic deprivation indices: the Townsend Deprivation Index, Social Deprivation Index, and Social Vulnerability Index. Additionally, we have included a brief description of how scores for each index were derived, highlighting the methodologies and calculation processes. These updates aim to clarify the construction of these indices and acknowledge that different methods can yield varying results. The following lines have been revised and added to the manuscript:

Page 6 Line 138-139

“County-level Townsend Deprivation Index scores were manually created following the methodology outlined in the original publication. [38]”

Page 7 Lines 144-152

“For the Social Deprivation Index, a composite measure was created using demographic variables from the ACS. These variables were standardized and weighted based on factor analysis to include only those strongly associated with deprivation. [30, 31] To construct the Social Vulnerability Index, ACS variables were ranked for all United States census tracts, with each variable assigned a percentile rank. For most variables, higher ranks indicated greater vulnerability, except for per capita income, where higher values indicated le

---

## [Decision Letter · Decision Letter 1]

17 Sep 2024

PONE-D-24-10230R1Socioeconomic deprivation and suicide in Appalachia: The use of three socioeconomic deprivation indices to explain county-level suicide ratesPLOS ONE

Dear Dr. Caswell,

Thank you for submitting your manuscript to PLOS ONE. After careful consideration, we feel that it has merit but does not fully meet PLOS ONE’s publication criteria as it currently stands. Therefore, we invite you to submit a revised version of the manuscript that addresses the points raised during the review process.

We look forward to receiving your revised manuscript.

Kind regards,

Angela Mendes Freitas

Academic Editor

PLOS ONE

Additional Editor Comments:

Dear authors,

Following the reviewer's feedback, there are still some issues that need to be addressed before your manuscript can be accepted for publication. Please check the comments below.

Best regards,

Reviewers' comments:

Reviewer's Responses to Questions

**Comments to the Author**

1. If the authors have adequately addressed your comments raised in a previous round of review and you feel that this manuscript is now acceptable for publication, you may indicate that here to bypass the “Comments to the Author” section, enter your conflict of interest statement in the “Confidential to Editor” section, and submit your "Accept" recommendation.

Reviewer #1: (No Response)

2. Is the manuscript technically sound, and do the data support the conclusions?

Reviewer #1: Yes

3. Has the statistical analysis been performed appropriately and rigorously? 

Reviewer #1: Yes

4. Have the authors made all data underlying the findings in their manuscript fully available?

Reviewer #1: No

5. Is the manuscript presented in an intelligible fashion and written in standard English?

Reviewer #1: Yes

6. Review Comments to the Author

Reviewer #1: The authors have attempted to address all comments from the previous round of review. However, I believe that the following two points still require major revisions:

1) In the "Supporting information - Manuscript Data", the following file does not present the data: “tdisuicide_final_ruralclass.xlsx”. In addition, all files are missing the data dictionary.

2) The attempt to differentiate "poverty" and "socioeconomic deprivation" was not clear. The authors presented their own definitions, but it is important to present the references that guided the choice of these definitions. Furthermore, there is still inconsistent information in the text, for example between the following excerpts:

Page 4. Lines 80-84. "Poverty refers specifically to a lack of financial resources, while socioeconomic deprivation encompasses a wider range of hardships, such as limited access to education, substandard housing, inadequate transportation, and diminished social capital. Unlike poverty, which focuses on income, socioeconomic deprivation provides a more holistic view by capturing multiple dimensions of material and social disadvantage".

Page 4. Lines 88-94. "For example, socioeconomic deprivation has been assessed using readily available poverty measures such as unemployment, education level, and income. [13] Studies in the United States have predominantly relied on income to measure poverty. [18-21] However, poverty assessed solely by income only indirectly measures socioeconomic deprivation. Poverty is a complex construct comprised of many factors that income alone does not fully capture, such as lack of material resources and overall standard of living. [20] These standalone measures of poverty may not capture socioeconomic deprivation comprehensively and accurately. [22]".

7. PLOS authors have the option to publish the peer review history of their article (what does this mean?). If published, this will include your full peer review and any attached files.

Reviewer #1: No

---

## [Author Response · Author response to Decision Letter 1]

20 Sep 2024

1) Thank you for bringing this to our attention. We have thoroughly reviewed the files to ensure all data are complete and have included the variable codebook on the first sheet of each Excel document.

2) We agree that there is a need for clarity regarding the distinction between poverty and socioeconomic deprivation. In our study, "poverty" specifically refers to a lack of financial resources, typically measured by income levels. This definition aligns with the conventional use of poverty metrics [17]. On the other hand, "socioeconomic deprivation" is a broader construct that includes poverty but extends beyond it to encompass various dimensions of material and social disadvantage. These additional factors include limited access to education, substandard housing, inadequate transportation, and/or diminished social capital [18]. This broader perspective allows us to capture a more comprehensive view of poverty that goes beyond mere financial constraints.

The revised text now reads as: 

Page 4. Lines 82-88. “Poverty has been generally used to refer to a lack of financial resources, typically measured by income. [17] Socioeconomic deprivation, however, is a broader concept that includes poverty but also captures additional material and social disadvantages such as limited access to education, substandard housing, inadequate transportation, and/or diminished social capital. [18] Vulnerability, on the other hand, reflects an increased susceptibility to adverse outcomes, such as suicide, due to the cumulative effects of socioeconomic deprivation [17, 19].”

Pages 4-5. Lines 89-95. “Different measures have been used to assess socioeconomic deprivation while examining its association with suicide. For example, socioeconomic deprivation has been assessed using measures such as unemployment, education level, and income. [13] Studies in the United States have predominantly relied on income to measure poverty. [20-23] Consequently,, income as a standalone measure of poverty may not capture socioeconomic deprivation comprehensively and accurately. [24] By focusing solely on income, important contributors to social and material disadvantage are often overlooked, leading to an incomplete assessment of overall socioeconomic deprivation.” 

Newly Added References:

17 Ouoya Z. Poverty And Vulnerability To Poverty: Conceptual Overview, Measurements And Causes. International Journal of Scientific and Research Publications (IJSRP). 2021;11:378-94.

18 Gweshengwe B, Hassan NH. Defining the characteristics of poverty and their implications for poverty analysis. Cogent Social Sciences. 2020;6(1):1768669.

---

## [Editor Report · Decision Letter 2]

7 Oct 2024

Socioeconomic deprivation and suicide in Appalachia: The use of three socioeconomic deprivation indices to explain county-level suicide rates

PONE-D-24-10230R2

Dear Dr. Caswell,

We’re pleased to inform you that your manuscript has been judged scientifically suitable for publication and will be formally accepted for publication once it meets all outstanding technical requirements.

Kind regards,

Angela Mendes Freitas

Academic Editor

PLOS ONE
---

## [Editor Report · Acceptance letter]

6 Nov 2024

PONE-D-24-10230R2 

PLOS ONE

Dear Dr. Caswell, 

I'm pleased to inform you that your manuscript has been deemed suitable for publication in PLOS ONE. Congratulations! Your manuscript is now being handed over to our production team.

Kind regards, 

on behalf of

Dr. Angela Mendes Freitas 

Academic Editor

PLOS ONE